# TFEB Signalling-Related MicroRNAs and Autophagy

**DOI:** 10.3390/biom11070985

**Published:** 2021-07-04

**Authors:** Davide Corà, Federico Bussolino, Gabriella Doronzo

**Affiliations:** 1Department of Translational Medicine, Piemonte Orientale University, 28100 Novara, Italy; davide.cora@uniupo.it; 2Center for Translational Research on Autoimmune and Allergic Diseases—CAAD, 28100 Novara, Italy; 3Department of Oncology, University of Torino, 10060 Candiolo, Italy; 4Candiolo Cancer Institute-IRCCS-FPO, Laboratory of Vascular Oncology, 10060 Candiolo, Italy

**Keywords:** TFEB, miRNA, autophagy

## Abstract

The oncogenic Transcription Factor EB (TFEB), a member of MITF-TFE family, is known to be the most important regulator of the transcription of genes responsible for the control of lysosomal biogenesis and functions, autophagy, and vesicles flux. TFEB activation occurs in response to stress factors such as nutrient and growth factor deficiency, hypoxia, lysosomal stress, and mitochondrial damage. To reach the final functional status, TFEB is regulated in multimodal ways, including transcriptional rate, post-transcriptional regulation, and post-translational modifications. Post-transcriptional regulation is in part mediated by miRNAs. miRNAs have been linked to many cellular processes involved both in physiology and pathology, such as cell migration, proliferation, differentiation, and apoptosis. miRNAs also play a significant role in autophagy, which exerts a crucial role in cell behaviour during stress or survival responses. In particular, several miRNAs directly recognise TFEB transcript or indirectly regulate its function by targeting accessory molecules or enzymes involved in its post-translational modifications. Moreover, the transcriptional programs triggered by TFEB may be influenced by the miRNA-mediated regulation of TFEB targets. Finally, recent important studies indicate that the transcription of many miRNAs is regulated by TFEB itself. In this review, we describe the interplay between miRNAs with TFEB and focus on how these types of crosstalk affect TFEB activation and cellular functions.

## 1. Introduction

Autophagy is a catabolic process decisive for the control of multiple biological functions in different cell types. It is involved in molecular trafficking, recycling of intracellular elements, distribution of nutrients during starvation, and control of organelle and protein amount through the removal of damaged or unnecessary ones [1]. A dysregulation of autophagy is associated with many diseases, including storage diseases, neurodegenerative disorders, cancer, microbial pathogenesis, inflammation, immunity alteration, muscular disorders, and heart diseases [2].

The oncogenic transcriptional factor EB (TFEB) is the most important transcriptional regulator of the lysosomal–autophagic pathway. TFEB, a member of the microphthalmia-transcription factor (MiTF)/TFE family, is highly conserved during the evolution [3,4,5,6,7,8] and is expressed in multiple cell types [3,4,5,6,7,8]. The transcription of human *TFEB*, located on chromosome 6 (6p21.1), generates an mRNA characterised by two non-coding and eight coding exons. Exon 3 contains the 5′ UTR followed by a start codon, while exon 10 presents a stop codon and the 3′ UTR. Different TFEB isoforms have been described and are characterised by the absence or the presence of 5′ exon (Figure 1) [9]. TFEB and the other proteins of MITF family bind to E-box (CANNTG) and a palindromic consensus sequence named coordinated lysosomal expression and regulation (CLEAR) motif CLEAR-box (GTCACGTGAC) elements in the promoter regions of their target genes [10,11,12]. This motif was clearly described in HeLa and endothelial cells [11,12,13,14]. ChIP-seq analysis of TFEB protein interactors profiling TFEB-mediated transcriptional regulation and genome-wide mapping of TFEB target sites and co-expression analyses [11,12] revealed that TFEB interactors can be associated in different gene categories involved in autophagy and lysosomal activities and biogenesis, as detailed above [11,12,13,14,15,16]. TFEB coordinates the expression of approximately 60 lysosomal hydrolases, as well as lysosomal and autophagosome membrane and accessory proteins [11,12,13,14,15,16], but also regulates vesicles formation, localisation, and flux [11,12,13,14,15,16]. Interestingly, numerous genes have related TFEB to the modulation of endoplasmic reticulum (ER) stress, as well as cellular metabolism [17,18], cellular proliferation, DNA replication, and cytoskeletal regulation [12]. In vitro and vivo studies have suggested TFEB as an important regulator of various cellular physiological and pathological processes in addiction to autophagy. TFEB supports metabolism, immunity, angiogenesis, inflammation, and cancer development [15,16,17,18,19,20,21].

TFEB activation occurs in response to multiple stress factors such as nutrient and growth factors alterations, hypoxia, and mitochondrial stress. The transcription factor action results from its cytosol-nuclear translocation mediated by a post-translational regulation by protein modifications and protein–protein interactions. Moreover, the cellular and nuclear amount of TFEB is also important: a transcriptional and post-transcriptional TFEB modulation is increasingly evident [11,12,13,14,15,16]. Multiple pathways intertwine, resulting in TFEB phosphorylation at residues S122, S142, and S211, which address its sub-cellular localisation (Figure 2) [11,12,13,14,15,16].

Different pieces of evidence support the fact that post-translational, transcriptional, and post-transcriptional regulation of TFEB are in part mediated by microRNAs (miRNAs). Moreover, miRNAs are able to control TFEB effects via the regulation of its target genes (Figure 3).

## 2. miRNAs and TFEB

miRNAs are small and single-stranded endogenous RNA molecules encoded by eukaryotic nuclear DNA. They mainly repress the translation or reduce the stability of target mRNAs to fine tune gene expression [22]. miRNAs are transcribed from coding and non-coding sections of genome, and they are indicated as regulatory RNAs [22,23,24]. Their DNA is transcribed by RNA polymerase II into long polyadenylated primary miRNAs (pri-miRNAs) that are then cleaved by the RNase III enzyme Drosha into miRNA precursors (pre-miRNAs), characterised by a hairpin-like secondary structure. Pre-miRNAs translocate in the cytoplasm through a direct interaction with Exp5 and are processed by the Dicer RNase enzyme. Dicer enzyme cuts the pre-miRNA to a mature length generating a doublestranded molecule that then via the RNA-induced silencing complex (RISC) can be unwound, and mature miRNAs can be generated. Mature miRNAs are composed by ≈19–25 nucleotides derived from the 5′ and/or 3′ arms of the precursor duplex, and therefore referred to as miRNA-5p and -3p, respectively. miRNAs reduce the expression of their target genes by binding a so-called miRNA “seed”, usually into 3-untranslated region of the target gene (3/-UTR). Rarely, they are also mapped at the mRNA 5′-UTR and coding sequence region [22,23,24]. One miRNA has the capability to modulate many target genes simultaneously and differently.

miRNAs are involved in the control of processes such as oxidative stress [25], cell proliferation [12,26], apoptosis [27,28], migration [29], and autophagy [30,31]. Cellular context and environmental conditions affect the expression of miRNAs and could modify their effects. The failure of miRNA regulation supports many diseases. In particular, dysregulation of autophagy-related miRNAs is associated with cancer, neurodegeneration, cardiovascular disease, and infections [30].

Recent findings demonstrate that miRNAs control TFEB activation, expression, and activity, but also the amount of the key genes of autophagic machinery. TFEB activity is carefully controlled transcriptionally, by post-translational modifications and protein–protein interactions [11,12,13,14,15,16], and many miRNAs further modulate these regulatory mechanisms (Figure 3). In particular, TFEB is a direct target of several miRNAs or these can indirectly modulate the transcription factor by acting on its post-translational activators or inhibitors [30,31]. In addition, several miRNAs are known to reduce the amount of mRNAs coding autophagy proteins [30,31]. These miRNAs are able to block all specific steps of the mechanism of autophagy coordinated or not coordinated by TFEB and in particular the early stage of autophagic vesicles formation [30,31]. Moreover, miRNAs tune the transcription of TFEB induced CLEAR genes [30,31]. Although less known, recent important studies indicate the involvement of TFEB in the transcription of specific miRNAs [12,32].

## 3. Post-Transcriptional Regulation of TFEB via miRNAs

Transcription factors are able to control the amounts of their target gene products according to the needs of the cell. Furthermore, transcription factors themselves are fine regulated and different studies suggested the existence of complex networks between they and miRNAs [23,24]. Different miRNAs are involved in the post-transcriptional regulation of TFEB as summarised in Figure 4. 

### 3.1. miRNA-128

miRNA-128 is a intronic miRNA encoded by miRNA-128-1 and miRNA-128-2 genes embedded in the introns of R3HDM1 (R3H domain-containing 1) and RCS (ARPP-21, cyclic AMP-regulated phosphoprotein) genes located on human chromosome 2q21.3 and 3p22.3, respectively. miRNA-128 tunes genes involved in cell proliferation, differentiation, migration, apoptosis, and survival, and it participates in the development of nervous system, musculoskeletal diseases, tumours, and angiogenesis [33,34,35].

Different studies predicted and validated TFEB as a target of the miRNA-128 [13,36,37]. In HeLa cells, TFEB mRNA degradation mediated by miRNA-128 leads to the downregulation of many CLEAR genes involved in lysosomal pathway [13]. A miRNA-128–TFEB interplay dysregulation was suggested as a trigger event of the alteration of the autophagy-lysosomal pathway observed in patients with Alzheimer’s disease (AD) compared to healthy subjects [37]. In particular, a bioinformatics analysis with “microRNA.org software” [37] has suggested that TFEB is a miRNA-128 direct target, and two potential binding sites were evidenced on *TFEB*. As previously described in the central nervous system [38], miRNA-128 amount is increased in cells of the peripheral system derived from patients with AD compared to healthy subjects. On the contrary, TFEB transcription and protein expression are decreased in monocytes and in lymphocytes from patients with AD compared with cells isolated from healthy subjects. ChIP assay demonstrated a reduced TFEB protein binding in nuclear regions in both cell types from AD patients compared with healthy subjects attributable to a decreased nuclear protein expression. Moreover, it has been reported that miRNA-128 upregulation supports the block of lysosomal–autophagic functions through the inhibition of lysosomal *Cathepsin B*, *D*, and *S* genes characterised by the CLEAR sequence in their promoter region [13].

### 3.2. miRNA-29

The miRNA-29 family includes miRNA-29a, miRNA-29b-1, miRNA-29b-2, and miRNA-29c encoded by genes located on chromosome 7 (7q32.3) and on chromosome 1 (1q32.2). The miRNA-29 family controls cell apoptosis, proliferation, and differentiation and exerts an antifibrotic activity targeting genes concerned extracellular matrix remodelling [39].

miRNA-29 acts as a tumour suppressor and therefore is frequently down-regulated in cancer cells [39]. In particular, it has been reported that pancreatic ductal adenocarcinoma (PDAC) cell lines are characterised by a reduced amount of miRNA-29 and by an upregulation of autophagy and in particular of TFEB and ATG9A [40], which is a transmembrane protein that supports the formation of autophagosomes [11,12,13,14,15,16]. Interestingly, the restoration of miRNA-29 expression blocks the autophagy pathway by inhibiting the protein expression of TFEB and ATG9A [39]. Different prediction algorithms suggested that both *TFEB* and *ATG9A* contain phylogenetically conserved miRNA-29 binding sites in their 3′ UTRs, and in vitro studies clearly supported the post-transcriptional regulation of TFEB and ATG9A expression by miRNA-29. Accordingly, miRNA-29 overexpression leads to an increased accumulation of autophagosomes and lysosomes and a decreased autophagosome-lysosome fusion [40]. A role of miRNA-29 in the control of autophagy and TFEB was also indicated in choroid/retinal pigment epithelial cells (RPE) [41]. In these cells, miRNA-29 overexpression leads to the reduction of cytosolic TFEB protein amount, but it does not modify its nuclear fraction, suggesting that the RPE autophagic pathway is likely TFEB-independent [41].

### 3.3. miRNA-300

A recent study identified a correlation between miRNAs and genes potentially associated with myocardial infarction. The authors evaluated the interplay between miRNA and the expression and function of target genes and assigned a diagnostic value of identified interactions [42]. TFEB is expressed in adult myocardium [43], and its expression is modulated during myocardial infarction. Interestingly, a correlation between TFEB and miRNA-300-3p was described. miRNA-330-3p is downregulated, while TFEB is upregulated during myocardial infarction [42].

### 3.4. miRNA-30

The majority of miRNAs reside in the cytoplasm and act as posttranscriptional regulators. However, some miRNAs are located in the nucleus and are involved in the transcriptional gene activation [44].

The miRNA-30 family is composed of miRNA-30a, miRNA-30b, miRNA-30c (c1 and c2), miRNA-30d, and miRNA-30e encoded by six genes located on human chromosomes 1, 6, and 8 [45]. These miRNAs target different genes and have multiple functions since they are characterised by the same seed sequence, but they differ in compensatory sequences in proximity of the 3′ end [46]. Nuclear miRNA-30b-5p is involved in the regulation of TFEB activity in human embryonic kidney epithelial cells [46]. In fact, it regulates genes through posttranscriptional regulation in the cytoplasm but also it moves into the nucleus where bound CLEAR elements blocking the transcriptional autophagic–lysosomal program induced by TFEB both in vitro and in vivo. miRNA-30b-5p presents a motif that is complementary to the promoter of these genes. miRNA-30b-5p does not repeal TFEB nuclear translocation or its protein synthesis but only inhibited the transcription of CLEAR genes [46]. The effects of miRNA-30b-5p on lysosomal biogenesis and autophagy was also evaluated in mouse liver in which autophagy was induced by 48 h fasting. The overexpression of mature miRNA-30b-5p via recombinant adeno-associated virus liver-specific results in the decrease of enrichment of TFEB and the inhibition of lysosomal and autophagic biogenesis genes [46].

**Figure 4 biomolecules-11-00985-f004:**
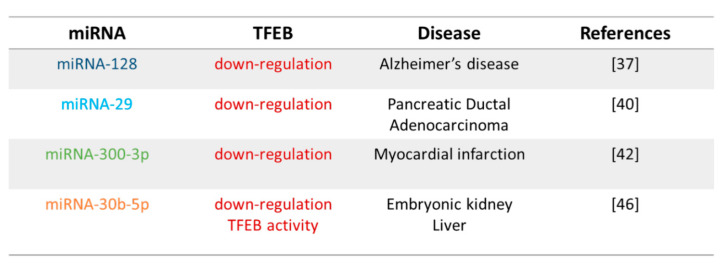
Post-transcriptional regulation of TFEB by miRNA and correlated diseases.

## 4. miRNAs Control of TFEB Activation

Environmental cues play a crucial role in transcriptional regulation. Nucleo-cytoplasmic shuttling of transcription factors are finally controlled by nutrient level, stress conditions, and the modulation of gene expression programs. An important mechanism on the basis of subcellular localisation of transcription factors is their phosphorylation [47].

TFEB sub-cellular localisation and the resulting activity is strictly affected by post-translational modifications [11,12,13,14,15,16,48]. The most important mechanism that controls TFEB subcellular localisation involves its phosphorylation on specific serine residues [48] mediated by different kinases (Figure 2).

Under nutrient-rich conditions, TFEB is predominantly inactivated via its phosphorylation and retention in the cytosol. On the contrary, during starvation, oxidative stress or under conditions requiring lysosomal activation, TFEB phosphorylation is blocked, and the transcription factor is able to translocate to the nucleus where it induces the transcription of CLEAR genes [11,12,13,14,15,16,48]. The nutrient availability supports TFEB lysosome localisation through amino acid-regulated Rag GTPases. TFEB-Rag interactions block the transcription factor to the surface of the lysosomes, and Rags are able to give information on the number of amino acids to mammalian target of rapamycin (mTOR) [48]. TFEB nuclear translocation is also promoted by infection [49,50], bacterial phagocytosis [51], inflammation [52], physical exercise [18], mitochondrial damage [53], and ER stress [17].

In addition to the aforementioned mechanisms, some miRNAs are involved in TFEB phosphorylation and cellular localisation by controlling the kinase pathways (summarised in Figure 5).

### 4.1. miRNA-211 and mTOR

The serine/threonine kinase mTOR has a crucial role in TFEB phosphorylation [47]. mTOR form two complexes, rapamycin-sensitive mTORC1 (containing mTOR, RAPTOR, MLST8, PRAS40, and DEPTOR) and rapamycin-insensitive mTORC2 (containing of mTOR, RICTOR, SINl, MLST8, and DEPTOR). mTORC1 is linked to cell growth, proliferation, survival, protein synthesis, and autophagy, whereas the mTORC2 complex is primarily involved in cell shape and metabolism, but it can also indirectly modulate autophagy [54]. In particular, mTORC1 is responsive to nutritional status of the cells and by phosphorylating TFEB and proteins needed in the formation of autophagosome such as ULK-1 and ATG13 strongly directs autophagy development. In normal conditions, TFEB is linked with Rag GTPases at the lysosomal surface, and mTORC1 phosphorylates it on Ser122, Ser142, and Ser211 and promotes its interaction with 14-3-3 scaffold proteins, thus preventing its nuclear translocation. During starvation, mTORC1 and Rag GTPase are inhibited, and TFEB, free from 14-3-3 complex, translocates from the lysosomal membrane to the nucleus [11,12,13,14,15,16,48].

Accumulating evidence indicates an miRNA contribution to autophagic pathway regulation by targeting different steps of mTOR pathway. While some miRNAs target mTOR directly, others modulate mTOR signalling by targeting proteins involved in its signalling cascade [31,55]. Recently, by exploiting an unbiased screen, different miRNAs (Figure 6) [31,55] have been identified to be able to reduce the inhibitor effects of mTOR on autophagy. The same components of mTORC complex such as mLST8, DEPTOR, Tti/Tel2, RAPTOR, RICTOR, PRAS40, and mSIN1 or mTOR regulators such as RHEB, TSC1/2, and RAG are targets of different miRNAs [31,55].

In particular, the miRNA-211 encoded by the sixth intron of the TRP channel member TRPM1 gene (melastatin 1) has been reported to exert a regulatory role of the mTOR–TFEB axis [56,57]. A high positive correlation of miRNA-211–MITF expression was observed in the skin cutaneous melanoma, while a variable positive correlation was present in the pan-kidney cohort testicular germ cell tumours, glioma, and ovarian serous cystadenocarcinoma [57]. In particular, in melanoma, in stressed condition, miRNA-211 binds RICTOR and inhibits mTORC1 complex through AKT. miRNA-211 overexpression decreases AKT phosphorylation at Ser473, which is a direct target of the MTORC2-associated MTOR Ser/Thr kinase. Parallel, miRNA-211 increase inhibits the activation of MTORC1 by AKT via MTOR Ser2448 phosphorylation [57]. Different bioinformatic tools have identified *RICTOR* as a potential direct target of miRNA-211. A miRNA-211-binding site in the 3ʹ UTR of the *RICTOR* has been mapped [57]. The overexpression of miRNA-211 and the block of mTORC1 permit the TFEB translocation to the nucleus, and the consequent autophagosome/autolysosome formation and protein degradation [57].

### 4.2. miRNA-21 and PTEN

AKT is likely to be involved in TFEB phosphorylation independently from mTOR. In particular, the inhibition of either PI3K or AKT result in TFEB activation and nuclear translocation similar to mTORC1 inhibition [48].

miRNA-21 is located in the 10th intron of the vacuole membrane protein-1 (VMP1) gene, which is involved in the formation of autophagosome. miRNA-21 is an oncogenic miRNA upregulated in many tumours where it controls cell proliferation, apoptosis, invasion, and metastasis, and supports drug resistance [58]. A fusion transcript of VMP1–miRNA-21 was described, suggesting that its transcription is regulated by its own promoter and by VMP1 promoter [58]. Of note, in colorectal cancer (CRC), PTEN gene was found to be a miRNA-21 target gene. PTEN can act as an upstream inhibitor of AKT, and it was demonstrated that miRNA-21 increases AKT phosphorylation by inhibiting the expression of PTEN [59]. Recent findings reported that TFEB nuclear translocation is increased in miRNA-21 null CRC cells, and this effect is abrogated by PTEN inhibitor, suggesting that miRNA-21 could inhibit the nuclear translocation of TFEB via PTEN/AKT [59].

### 4.3. miRNA-33 and AMPK

The serine/threonine protein kinase adenylate-activated protein kinase (AMPK) is also involved in TFEB post-translational modulation [11,12,13,14,15,16,48]. AMPK supports different signalling pathways that are involved in the upregulation of catabolic cellular processes and the downregulation of anabolic processes [60]. Under physiological or pathological conditions, during cell starvation situation or when cellular energy levels are low, AMPK is activated by multiple phosphorylation by different upstream kinases [60]. AMPK tunes autophagy via a negative regulation of mTORC1 [61,62] or via the phosphorylation of ULK-1, involved in autophagy initiation, supporting autophagy biogenesis [62]. In particular, AMPK alone or co-opting mTORC1 pathway supports the nuclear translocation and activation of TFEB [48,63,64]. Moreover, AMPK mediates phosphorylation of TFEB in S466, S467, and S469 residues, leading to TFEB transcriptional activity [64].

miRNA-1265 [65], miRNA-451 [66], miRNA-21 [67], and miRNA-519d [68] are able to control AMPK and AMPK-mTOR signalling in autophagy context. Moreover, miRNA-33 represses multiple genes related to autophagy and in particular is involved in the inhibition of AMPK mediated activation of TFEB [69] (Figure 2).

miRNA-33 is an intronic miRNA and is characterised by two isoforms, miRNA-33a and miRNA-33b. These miRNAs respectively derive from intron 16 of *SREBP2* gene on chromosome 22 and intron 17 of *SREBP1* gene on chromosome 17, and their expression is under the control of their host genes. In silico prediction and molecular data indicate *PRKAA1* and *PRKAA2*, which encode AMPKα proteins [70], as robust targets of miRNA-33 [70]. Via miRNA-33, mycobacterium tuberculosis supports its survival inhibiting TFEB activation and lysosome biogenesis and promoting the accumulation of lipid body source of nutrients [69]. Although TFEB gene lacks a specific binding site for miRNA-33 in 3′ UTR, miRNA-33 overexpression reduces TFEB and CLEAR gene transcription, and miRNA-33 silencing increases TFEB protein expression and its nuclear localisation. Of note, the cell silencing of AMPK abrogates the miRNA-33 effects on TFEB and CLEAR genes, suggesting a miRNA-33 AMPK-mediated effect [69].

### 4.4. miRNA-211 and Ezrin

TFEB phosphorylation and nuclear translocation is also mediated by a Ca^2+^ signalling pathway. In particular, the release of lysosomal Ca^2+^ via channel mucopilin 1 (TRPML1) supports “calcium microdomains” that induce the activity of calreticulin phosphatase (PPP3CB) [71]. Calreticulin binds and dephosphorylates TFEB at S211 and S142, thus inducing its nuclear translocation. Moreover, cytosolic Ca^2+^ decreases mTORC1 activity and TFEB phosphorylation. Both these events result in the nuclear translocation of TFEB and in the transcriptional induction of CLEAR genes. It was described that Ezrin, a cytoskeleton associated protein, regulates Ca^2+^ homeostasis maintaining ion channels at plasma membrane [71]. The inhibition of Ezrin supports TFEB dephosphorylation and its nuclear translocation.

In retinal pigment epithelium, miRNA-211 directly represses Ezrin, supporting the Ca^2+^ calcineurin pathway and therefore TFEB nuclear translocation and autophagy. On the contrary, the silencing of miRNA-211 in RPE cells inhibits the light-mediated induction of lysosomal biogenesis and results in severely compromised vision [72].

**Figure 5 biomolecules-11-00985-f005:**
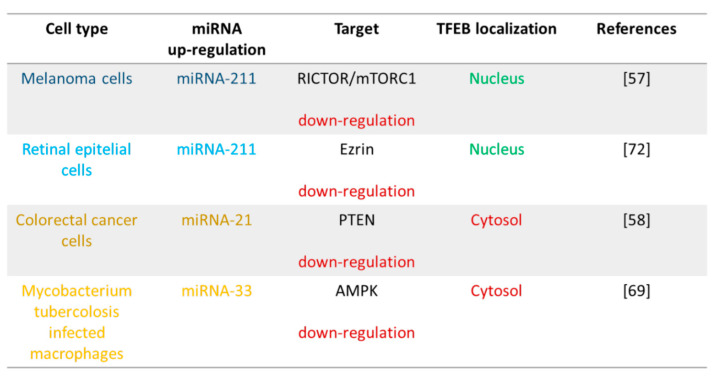
Regulation of TFEB activation by miRNA.

## 5. Autophagic Genes and miRNAs

Autophagy is a complex cellular process that comprises different steps [73]. The activation or inhibition of the autophagy and in particular its progression is based on the transcription and expression of many proteins that are involved in autophagic initiation, vesicle nucleation, elongation, maturation, and lysosomal fusion. Many of these proteins are TFEB targets and contain or the palindromic consensus sequence CLEAR or a partial CLEAR motif (TCACG) on their promoters [11,12,13,14,15,16,48].

As summarised in Figure 6, many autophagy molecules and in particular CLEAR molecules are miRNA targets [30,31,74,75]. Moreover, a specific miRNA can inhibit or activate different molecules at the same time increasing the complexity of the regulatory system, thus resulting in a tricky network of interacting elements. 

**Figure 6 biomolecules-11-00985-f006:**
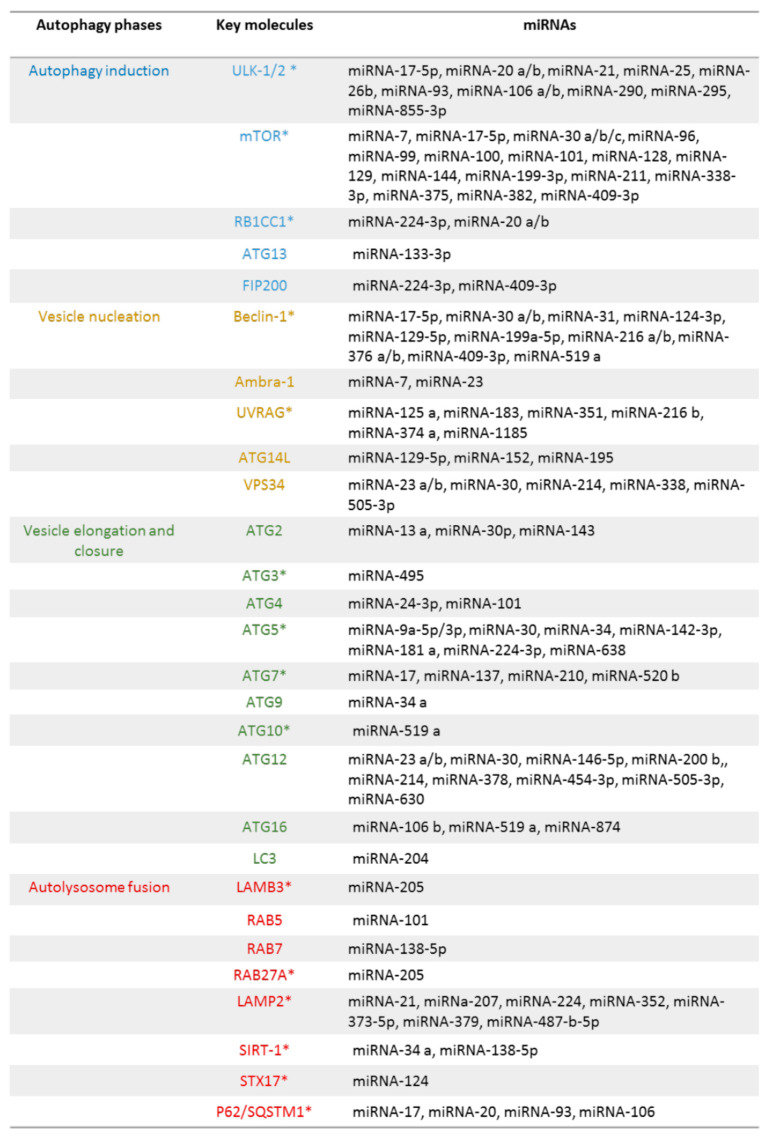
Regulation of outophagy by miRNA.Genes marked with * are TFEB target genes [30,31,74,75].

### 5.1. Autophagy initiation

Autophagy initiation which consists of the formation of phagophore, is supported by unc-51-like kinase (ULK) complex, consisting of ULK-1/2, autophagy-related gene (ATG)-13, ATG101, and FAK family kinase-interacting protein (FIP200) [73]. ULK-1/2 but also the other elements of this complex are affected by miRNAs. ULK-1 expression is regulated by miRNA-17 family and by a large number of other miRNAs including miRNA-20a, miRNA-106b, miRNA-93, miRNA-17-5p, miRNA-26a, miRNA-489, miRNA-142-5p, and miRNA-25. ULK-2 is a target of miRNA-26b and miRNA-885-3p. The 3′ UTR of ATG13 and FIP200 is directly targeted by miRNA-133a-3p, and miRNA-20a, miRNA-20b, miRNA-224-3p, and miRNA-309-3p, respectively [30,31,74,75].

### 5.2. Vesicle Nucleation

Vesicle nucleation, leading to the formation of autophagosomes, is based on the activity of a complex constituted by PI3KC3 (hVPS34), BECLIN-1, p150, and ATG14L, and requires other regulatory proteins such as AMBRA1, BIF-1, UVRAG, and RUBICON [73]. Many studies have focused attention on BECLIN-1 miRNAs and suggested that miRNA-30 family, miRNA-124-3p, miRNA-216b, miRNA-17-5p, and miRNA-376b and miRNA-143 are involved in the control of its expression. miRNA-23a and miRNA-7 control AMBRA-1, while miRNA-125 family, miRNA-183, miRNA-216b, miRNA-351, miRNA-374a, miRNA-630, and miRNA-1185 are related to UVRAG [30,31,74,75].

### 5.3. Vesicle Elongation

Vesicle elongation provides ubiquitination through the ATG12-5-16 and the ATG8/LC3-lipid conjugation system, two ubiquitination-like conjugation pathways [73]. Many proteins and corresponding miRNAs participate in this phase. ATG12 is modulated by miRNA-30 family members such as miRNA-23a, miRNA-23b, miRNA-214, and miRNA-505-3p. The miRNA-30 family is also involved in the regulation of ATG5, which is further controlled by miRNA-9a-5p, miRNA-142-3p, miRNA-181a, miRNA-224-3p, and miRNA-638. ATG16L1 is modulated by miRNA-142-3p, miRNA-17 family members, miRNA-106a, and miRNA-106b, while ATG7 is controlled by miRNA-17 family members, miRNA-137, miRNA-210, and miRNA-520b. The 3′ UTR of ATG10 mRNA is targeted by miRNA-4458, miRNA-4667-5p, and miRNA-4668-5p, and that of ATG4D is regulated by miRNA-101. ATG9, a transmembrane protein, is involved in vesicle maturation, and miRNA-34a controls its cellular levels [30,31,74,75].

### 5.4. Autolysosome Fusion

At the end of autophagic machinery, autophagosome membrane blends with lysosome giving origin to autophagolysosome. The contents of autophagosomes are exposed to the lysosome enzymes and are thus degraded [73]. Molecules regulating autophagolysosome assembly are RAB7, LAMP1, and LAMP2. RAB7 indirectly via SIRT1 is modulated by miRNA-138-5p, while miRNA-487b-5p, miRNA-207, miRNA-352, miRNA-21, miRNA-224, miRNA-373-5p, and miRNA-379 control LAMP-2. miRNA-205 has as a target RAB27A and LAMP3, also involved in this phase of autophagy. miRNA-33a-5p and miRNA-33a-3p have several inhibitory effects: they modulate ATG5, ATG12, and LC3B, but in parallel inhibit AMPK-autophagic activation and TFEB and FOXO3 transcriptional activity [30,31,74,75].

## 6. TFEB Regulation of miRNA Expression

As previously described [22,23,24], the “transcriptional events” are the result of a complex circuit that involves the interaction of transcription factors, transcription activators or suppressors, long-non coding RNAs, and miRNAs.

TFEB is not only modulated by miRNAs, but also is able to induce their expression by acting on the promoter of pri-miRNAs (Figure 7). The TFEB final cellular effects are the result of a balance of the directed interplay of the transcription factor with the target gene promoters and the modulation of the amount of these transcripts through a feedback mechanism mediated by TFEB-induced miRNAs expression.

### 6.1. TFEB and IRS: miRNA-335-3p, miRNA-495-3p, and miRNA-548o-3p

A clear example of this complex regulatory circuits relies on some activity of TFEB in the regulation of the vascular response to ischemic injury and of vascular development. Recently, different studies have suggested a role of TFEB in the control of vascularisation in placenta, in embryo, and in adult mouse development [5,12], and post-ischemic angiogenesis is in part mediated by the control of endothelial cell proliferation [76,77]. Moreover, a role of the transcription factor in endothelial inflammation [76,77] and metabolism [32] has been suggested.

In this frame, some of the biological effects of TFEB and in particular some of its vascular actions are supported by the intermediation of miRNAs.

Recently a small RNA-seq analysis on human coronary artery endothelial cells identified 176 small RNAs differentially expressed after TFEB overexpression, of which 65.5% are miRNAs and 3.4% are predicted miRNAs, 17.5% are Piwi-interacting RNA (piRNAs), 8.5% are small nucleolar RNA (snoRNAs), 3.4% are small nuclear RNAs (snRNAs), and 1.7% are ribosomal RNAs (rRNAs) [32].

Among the 122 miRNAs differentially modulated by TFEB, miRNA-216-3p, miRNA-21-5p, Let-7i-5p (microRNA lethal-7i-5p), and miRNA-486-5p are the most expressed [32] (Figure 7). In particular, different prediction databases have suggested nine candidate miRNAs regulated by TFEB as modulator of *insulin receptor substrate* 1 (*IRS1*) [32]. Among them, miRNA-335-3p, miRNA-495-3p, and miRNA-548o-3p expression is inhibited by TFEB overexpression. Little is known about miRNA-548o-3p, while miRNA-335-5p and miRNA-495-3p are related to tumour development regulating cell proliferation, apoptosis, hypoxia resistance, migration, epithelial–mesenchymal transition, invasion, lymph node metastasis, and chemotherapy resistance [78,79]. In particular, miRDB prediction suggests different binding sites of miRNA-495-3p, miRNA-335-3p, and miRNA-548o-3p in the IRS1 3′ UTR. TFEB inhibition increases IRS1 protein expression while the cell treatment with miRNA mimics significantly reverts TFEB’s activities [32]. Endothelium-*Tfeb* knockout mice and endothelium-*Tfeb* transgenic mice suggested that TFEB via miRNA-IRS pathway is able to control glucose tolerance under high-fat diet conditions. TFEB modulates endothelium metabolism: it is able to upregulate IRS1 and in this way AKT signalling and glucose uptake in endothelial cells [32].

**Figure 7 biomolecules-11-00985-f007:**
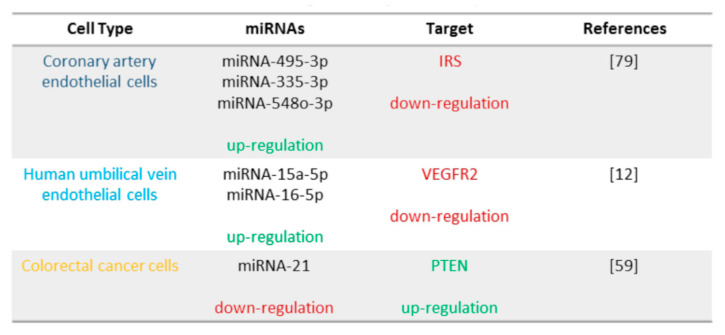
TFEB regulation of miRNA expression.

### 6.2. TFEB and VEGFR2: miRNA-15a-5p and miRNA-16-5p

TFEB is involved in the development of vasculature in embryo and new-born mice. It directly modulates the promoter of *cyclin-dependent kinase 4* (*CDK4*), a key molecule of G1-S phase of cell cycle transition, and indirectly via miRNA controls the quantity of vascular endothelial growth factor (VEGF) receptor (R)-2, the most important endothelial receptor [12]. It has been observed that TFEB silencing leads to the upregulation of VEGFR2 transcript and protein expression, but these phenomena are independent from a direct binding of TFEB on *VEGFR2* promoter.

The intersection of published data about the genomic location of angiogenic miRNAs [80,81] and ChIP-seq dataset on TFEB overexpressing endothelial cells, suggests the “*deleted in leukemia-2*” (*DLEU2*) and the “structural maintenance of chromosome 4” (SMC4) genes as putative miRNA host genes involved in VEGFR2 regulation by TFEB (Figure 8). 

DLEU2 is a tumour suppressor gene frequently deleted in haematological malignancies [82], while SMC4 is implicated in maintaining chromosome stability and dynamics in different solid cancers [83]. DLEU2 and SMC4 host, respectively, the miRNA-15a/16-1 and the miRNA-15b/16-2 clusters. The miRNA-15a/16-1 cluster produces miRNA-15a-3p, miRNA-15a-5p, miRNA-16-3p, and miRNA-16-5p, whereas the miRNA-15b/16-2 cluster generates miRNA-15b-3p, miRNA-15-5p, miRNA-16-2-3p, and miRNA-16-2-5p (Figure 4) [84].

In endothelial cells, miRNA-15a-5p and miRNA-16-5p are expressed at high levels, whereas miRNA-15a-3p and miRNA-16-1-3p levels are negligible. ChIP-seq and ChIP-qPCR indicate a TFEB binding site in *DLEU2* but not in *SMC4* promoter. TFEB overexpression and silencing respectively up- and downregulate DLEU2, miRNA-15a-5p, and miRNA-16-5p, suggesting a TFEB positive control of their transcripts. On the contrary, VEGFR2 expression is specular with respect to TFEB and miRNA-15a-5p and miRNA-16-5p expression, suggesting an inhibitory loop [12].

This TFEB miRNA regulation was confirmed in vivo with a specific endothelium *Tfeb*-knockout mouse model. In embryo and new-born mice, *Tfeb* endothelial silencing supports defects in vascular development and in retina and kidney angiogenesis [12].

### 6.3. TFEB and PTEN: miRNA-21

Previously, works described that miRNA-21 inhibited the nuclear translocation of TFEB via PTEN/AKT [59]. miRNA-21 is hosted in *VMP1* gene, which is a TFEB target gene. VMP1 is involved in the formation of cell–cell contacts and tight junction supporting invasion and metastatic power of cancer cells. In particular, in CRC VMP1 is downregulated, supporting the inhibition of cell adhesion and the increase of their migration. Chip-seq analysis revealed that TFEB specifically binds *VMP1* promoter, and this effect is upregulated by starvation. In CRC cells, TFEB-VMP1 binding downregulates the transcription of mature miRNA-21, VMP1-miRNA-21, and pri-miRNA-21. Moreover, TFEB blocks the interplay between the transcription factor STAT3 and pri-miRNA-21 promoter. These data suggest an inhibitory feedback regulatory effect of miRNA-21 on VMP1 expression via a PTEN/AKT/TFEB pathway [59].

## 7. Conclusions and Perspectives

Many cellular mechanisms, including autophagy, are fine-tuned both at the transcriptional the post-transcriptional levels, with miRNAs being some of the most important regulators for this latter type of interaction. Their abundance, activity, and function correlate with the cellular biogenesis, and like the other intracellular substances, miRNA levels can also be controlled via autophagosome–lysosome degradation [30,31,74,75]. Cancer, inflammatory diseases, and cardiac and vascular pathologies find a major cause in the alteration of the miRNA–autophagic gene axis [30,31,74,75].

Transcription factors lie at the starting point of multiple signalling networks, and their dysregulation expression and control via enhancers and inhibitors contributes to the onset and progression of many human diseases including cancer, diabetes, inflammatory disorders, and cardiovascular disease. Numerous transcription factors are involved in the modulation of autophagy-related genes, among which, the most representative are TFEB, cAMP response element-binding protein (CREB), Forkhead box O proteins (FOXOs), farnesoid X receptor (FXR), and peroxisome proliferator-activated receptor alpha (PPARα) [85].

Transcription factors and miRNAs are known to act in composite gene regulatory circuits, in which their interplay supports complex regulatory patterns. Exerting this conceptual layout on the TFEB transcription factor, its overall effects are thus probably mediated by its fine regulation at post-transcriptional and post-translational levels, and miRNAs might be involved in one or more of these processes. Moreover, in feed-forward or feed-back control loops, TFEB-miRNAs might amplify or limit TFEB cellular functions.

In this review, we analysed literature data currently supporting the reciprocal interaction between TFEB and the miRNA-mediated post-transcriptional regulation layer.

In the first part, the regulation of TFEB itself by miRNA was investigated. TFEB appears to be a target of several miRNAs that can alter and fine-tune its final level of expression and protein. Further literature mining pointed out the close connection also between the elements capable of activating and deactivating TFEB through phosphorylation events and miRNAs. The final expression level of a certain transcription factor is pivotal for its precise action on the network of target genes, and therefore, in the case of TFEB, its post-transcriptional regulation exerted by miRNAs results in a significant impact on the network of CLEAR-responsive genes.

In the second part of the work, we explored the possible role of miRNAs with respect to autophagic genes known to be transcriptionally regulated by TFEB. Several autophagic genes appear to be targets of miRNAs, thus suggesting a major role of the miRNA-mediated post-transcriptional regulatory network in establishing the final expression level of the autophagic pathway.

In the third part, we dealt with the possible transcriptional regulation of some miRNAs by TFEB. Some of the data in the literature support the idea that TFEB can also transcriptionally regulate, in addition to protein coding genes, a large series of miRNAs. It is worth noting that a single miRNA is itself capable of regulating a large network of target genes, which can on the one hand overlap, on the other hand expand, the actual catalogue of TFEB-modulated genes, directly or indirectly regulated. The overall action of TFEB of a certain phenotype is therefore most likely due to the finely orchestrated sum of direct or indirect interactions, in which miRNA-mediated action plays a primary role.

miRNAs are involved in cancer cell survival, proliferation, metabolism, oxidative stress supporting cancer development, progression, vascularisation, and metastasis also via the modulation of autophagy-related genes [30,31,74,75]. miRNA alteration is observed in several diseases, and many miRNAs have nowadays become prognostic and/or predictive in particular for the management of cancer patients [86,87].

Abnormalities of TFEB expression, activation, and localisation can contribute to lysosomal or neuro-degenerative storage diseases; muscular, renal, and liver pathologies; atherosclerosis; immune and vascular disorders; and tumours [19].

In this review, we summarised an extensive network of interactions between miRNAs and TFEB. In particular, we emphasised how the miRNA post-transcriptional and post-translational regulation of TFEB is a key point in the control of tumours (i.e., colorectal and kidney cancer, pancreatic and ovarian adenocarcinoma, melanoma, and glioma) but also of myocardial infarction and liver functions. Currently, TFEB control of miRNA expression is involved in vessel development and endothelial metabolism. It should be stressed that miRNAs are key molecules in endothelium so much that their complete depletion suppress tumour angiogenesis [88].

For these reasons, the selective regulation of TFEB expression/activity also via miRNAs looks to be a promising therapeutic intervention strategy for several diseases correlated with autophagy, but not only these diseases.

In summary, this review emphasises a new aspect of TFEB control via miRNA and suggests a putative new design strategy for the transcription factor activity during onset and progression of diseases such as infection, neurodegeneration, cardiovascular diseases, aging, and cancer.

## Figures and Tables

**Figure 1 biomolecules-11-00985-f001:**
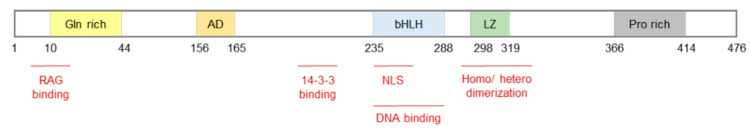
Representation of TFEB functional domains. TFEB contains N-term transcriptional activation domain (AD), basic helix-loop-helix region (bHLH), leucine zipper (LZ), and proline-rich domain (Pro-rich). Numbers indicate amino acid location in TFEB protein.

**Figure 2 biomolecules-11-00985-f002:**
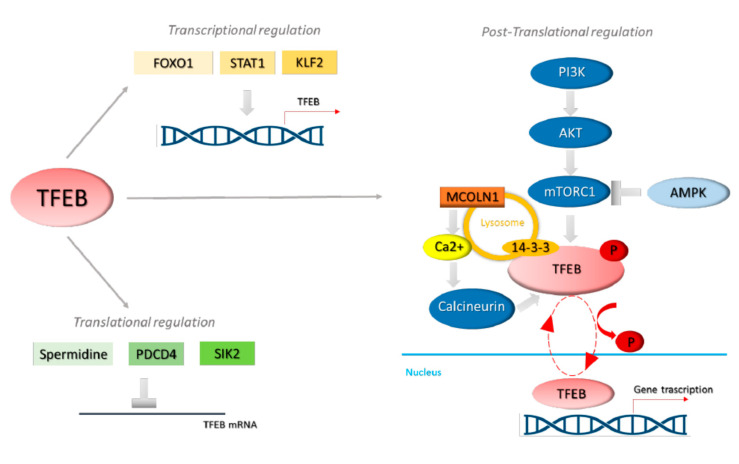
Control of TFEB transcription, translation, sub-cellular localisation, and activity. TFEB transcription level is controlled by the binding of its promoter by Forkhead box O1 (FOXO1), signal transducer and activator of transcription 1 (STAT1), and by Krüppel-like factor 2 (KLF2) [11,12,13,14,15,16]. TFEB translation is also regulated by programmed cell death 4 protein (PDCD4) and spermidine [11,12,13,14,15,16]. TFEB sub-cellular localisation and activity is under the control of phosphorylation level. In particular, under nutrient-rich conditions mTORC1 is activated via the PI3-K/Akt pathway. TFEB is phosphorylated by mTORC1 and retained in the cytoplasm on the lysosomal surface via the 14-3-3 proteins binding. In stress conditions, mTORC1 is inactivated, also via AMPK, and TFEB, no longer phosphorylated, is free to translocate to the nucleus. Moreover, in these conditions, Ca^2+^ is released from lysosome and induces calcineurin activation that dephosphorylates TFEB, supporting its nuclear translocation. TFEB cellular localisation is mediated by the phosphorylation in Ser sites. TFEB cytosolic retention is supported by its phosphorylation at (i) Ser122, Ser211, and Ser142 by mTOR; (ii) Ser134 and Ser138 by GSK3; and (iii) Ser142 by ERK-1/2. TFEB nuclear export is induced by its phosphorylation at (i) Ser 138 by GSK3; (ii) Ser142 by CDK4 or CDK6 [15,16,19].

**Figure 3 biomolecules-11-00985-f003:**
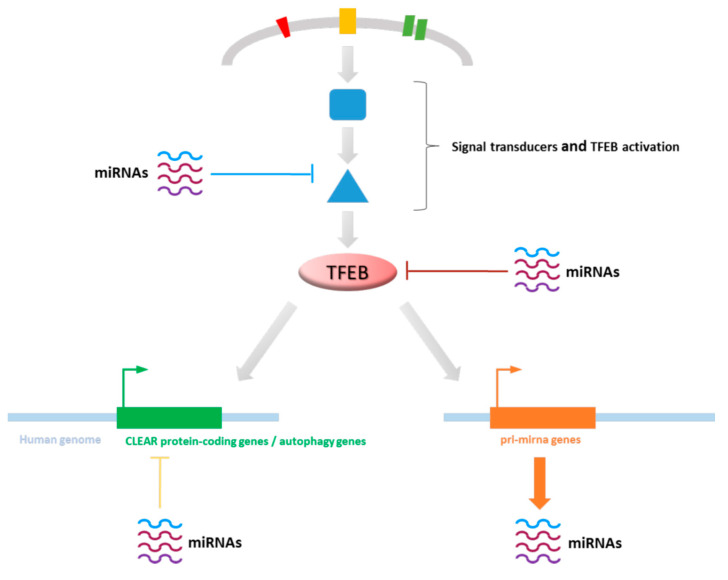
TFEB and miRNAs. TFEB exerts its function being embedded in a complex network of molecular species, regulated by miRNAs at various steps. Several miRNAs are known to regulate TFEB itself or its activation (see Figures 4 and 5). Among TFEB targets, several CLEAR responsive genes involved in autophagy are post-transcriptionally regulated by miRNAs (see Figure 6). Finally, TFEB can directly promote the transcription of miRNA precursors (see Figure 7).

**Figure 8 biomolecules-11-00985-f008:**
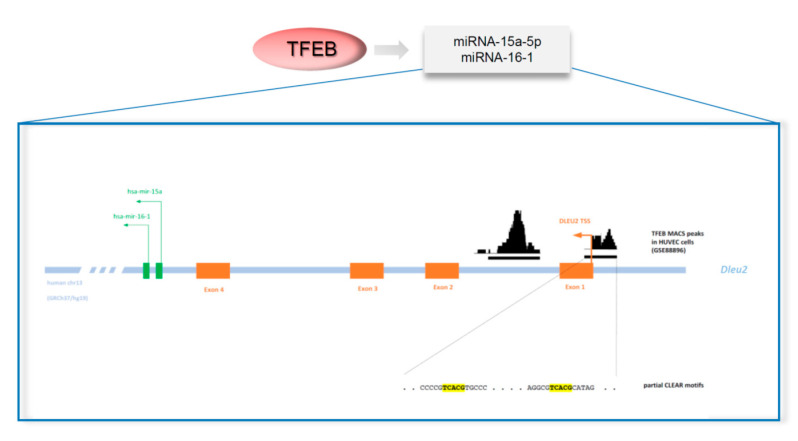
Example of TFEB binding sites, directly regulating the transcription of miRNAs. The non-coding human *DLEU2* gene locus is schematically represented (not in scale). The *DLEU2* gene hosts inside one of its introns the hairpin precursors hsa-mir-16-1 and hsa-mir-15a. In a model of human endothelial cells, data from GSE88896 [12] highlighted the presence of two TFEB MACS peaks in the close proximity of the *DLEU2* promoter region and the direct regulation of mature miR-15a-5p and miR-16-5p by TFEB. Inspection of the TFEB peaks sequences revealed the presence of multiple partial CLEAR binding sites.

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
