# Peer review of "TFEB Signalling-Related MicroRNAs and Autophagy"

_biomolecules, 2021, doi:10.3390/biom11070985_

Round 1
Reviewer 1 Report
The article summarises the findings in the literature that relates to the interactions between microRNAs and TFEB. It starts by presenting the role of TFEB in controlling autophagy and the biology of microRNAs. Then, the authors enumerate the microRNAs that regulate TFEB at different stages transcription, post-transcription, and activation. Next, the article ends by discussing the role of microRNAs in regulating other autophagy genes and the role of TFEB in regulating the expression of microRNAs. The "conclusion and perspectives" section summarizes the three different parts of the manuscript.
I believe this article can be of benefit for those who are interested in this topic. That said, I also believe there is room for improvement. The following are a few issues and suggestions that I hope would help in this regard.
- The article focuses on enumerating and discussing the studies that show one or more regulatory links between microRNAs and the transcription factor. While this is valuable in itself, It would be a great benefit to the reader if the authors also provide a synthesis of this body of evidence. That is also to show what motivates this kind of research, how these findings drive knowledge forward (of autophagy or disease), and what are the points of contention in this field.
- The structure of the article is fine but could be improved by adding subsections with headings.
- On reading the manuscript, I could not get a feel for the general direction (past and present) and the outstanding research questions. The section "Conclusion and perspectives" would be the perfect place for the authors to speculate on what questions this body of work succeeded in answering and what questions remain to be addressed.
- The authors should take advantage of graphical figures to explain some of the mechanistic detail that they think important to understand the topic.
- The existing graphics could be improved. I suggest, for example, reducing the outlining shapes, choosing a consistent color palette, and having a direction where the information flow.
- Tables can be a great tool to use to organize lists of entities by one or more categories. I think the information in Figure 2 would be better served as a table.
Author Response
June 16, 2021
To the Editor
of Biomolecules
Dear Editor,
I submit the revision of our MS entitled “TFEB signaling-related MicroRNAs and autophagy” written by myself, Davide Corà and Federico Bussolino, which was reviewed accordingly to referees’ comments.
Attached you read the point-to-point replies to the referee’s criticisms.
I hope now that this MS is suitable to be published on Biomolecules journal.
Looking forward to hearing from you,
Sincerely.
Gabriella Doronzo, PhD

Reviewer 2 Report
Dear authors,
The article as "TFEB signaling-related MicroRNAs and autophagy" is quite critical in explaining the role of TFEB in autophagy via miRNAs. However, when reading this article, I was confused by the way that authors expressed the importance of TFEB in autophagy. The data are so rich and need some clarification.
Major
1. The arrangement as first part, miRNAs regulate TFEB; second part, miRNAs regulate autophagy related genes under the TFEB control; third part, TFEB regulates miRNAs. The arrangement is good for the readers. However, the data are so huge, it would be not easy for the readers to know everything at a glance. Please consider to summarize in a table.
2. The miRNAs that mentioned in the figures or tables should also be mentioned in the text.
3. For readers, miRNA-TFEB-autophagy interaction is important. However, the diseases mediated by this axis would be more interesting for the readers. Please consider to add the diseases in the text and table.
Minor
1. There are several typos and grammar errors should be corrected. Please have someone who is mastering English writing for further proof.
2. The font should be carefully checked as line 140, ; miRNA-29 acts as; line 397, protein; etc
3. The arrangement of every paragraph should be carefully checked as line 457
Author Response

(The authors gave the same response as above.)

Reviewer 3 Report
Corà, Bussolino & Doronzo present here a detailed and through review of the regulation of TFEB (and to an extent, autophagy more generally) by microRNAs. The topic is very focused, and somewhat niche, but is certainly original and of interest. There approach is very thorough, and I think that this manuscript would make a worthy addition to the scientific literature. Before that, I have some issues regarding the presentation that should be addressed:
The authors go into extreme detail at various points of the manuscript. In some instances it is justified (e.g. listing components of mTORCs) because this detail is required to understand the role of microRNAs in modulating it. In other instances, it is not clear why it is included, e.g. why do they give details of exact residues that AMPK phosphorylates, when this is only vaguely related to the main topic of the review and no context is provided as to why these specific sites are important. The authors should therefore have a proof read of their content, to ensure that all detail included in it is relevant to the main topic. In some instances, they could maybe present it in a different format instead (e.g. a table of microRNAs that affect TFEB directly, which includes details of their loci and genes). Where data is displayed visually (notably figure 2), detail should not be repeated in the main text as well.
The introductory section, detailing the structure of the TFEB gene and protein, and their normal function, appears somewhat rushed, and would certainly benefit from a figure summarizing these structures visually.
The “Conclusions and perspectives” section is mainly a summary of the paper – a few more conclusions and perspectives would be nice!
The quality of the writing and English is very good, but a few formatting issues should be addressed:
- It would be good to consistently differentiate between genes/transcripts and proteins by writing gene/transcript names in italics/cursive (e.g. TFEB gene, TFEB protein).
- Where whole gene/protein names are given, if the first word is capitalized, then every major word should be capitalized (e.g. page 6: Vacuole Membrane Protein 1, page 10: Deleted in Leukemia)
- On page 10, please use periods/full stops, and not commas, to indicate decimals.
Page 2, Line 94: It is not clear whether “controlled transcriptionally” means control of the transcription of TFEB transcripts, or by the TFEB protein
Page 3l Line 10 (?): Software does not “reveal” microRNA targets, but can predict them.
Author Response

(The authors gave the same response as above.)

Round 2
Reviewer 1 Report
In their revision and reply, the authors addressed some of the concerns raised in the review by changing the text, graphics, and tables. These included adding a detailed discussion of the pathological implications of the reviewed studies on TFEB/microRNA interactions. The main changes were in the "conclusion and prospective" section. In addition, the authors slightly modified the graphics and transformed one of the figures into a table as suggested.
The authors didn't address some of the issues raised earlier. Namely
- The structure of the article is fine but could be improved by adding subsections with headings.
- The authors should take advantage of graphical figures to explain some of the mechanistic detail that they think important to understand the topic.
- In addition, the modified figures don't greatly improve on the previous versions. Therefore, careful consideration might be required to take full advantage of the graphic devices. Mainly choosing the appropriate points to illustrate and making the graphs visually appealing.
Author Response
Q. In their revision and reply, the authors addressed some of the concerns raised in the review by changing the text, graphics, and tables. These included adding a detailed discussion of the pathological implications of the reviewed studies on TFEB/microRNA interactions. The main changes were in the "conclusion and prospective" section. In addition, the authors slightly modified the graphics and transformed one of the figures into a table as suggested.
The authors didn't address some of the issues raised earlier. Namely
Q. The structure of the article is fine but could be improved by adding subsections with headings.
R. As suggested we introduced subsections to better point attention on the different points described in each paragraph.
Q. The authors should take advantage of graphical figures to explain some of the mechanistic detail that they think important to understand the topic.
Q. In addition, the modified figures don't greatly improve on the previous versions. Therefore, careful consideration might be required to take full advantage of the graphic devices. Mainly choosing the appropriate points to illustrate and making the graphs visually appealing.
R. I thank this reviewer for his/her suggestions. As suggested we modified figure 2 in which we summarized the topic concepts of TFEB transcription, translational and post-translational regulation. We have also inserted the figure 3 in which we have synthesized the interaction between TFEB and miRNAs. We preferred to summarize the other key concepts discussed in each paragraph schematically in the tables 1-4.
Reviewer 2 Report
Dear Authors,
I received this article as a PDF file. There are still some parts needed to be corrected before publication.
- Missing parts for Figure legend as Figure 1 which might be related to PDF file issue
- For a paragraph, it usually consisted of 5-10 sentences. However, there are still some paragraphs consisted of 1 or 2 sentences which might be not good for a reader. Please consider rearrangement such as line 303-315. Please check the paper submitted.
- Line 298, extra space
Author Response
Reviewer 2
Q. I received this article as a PDF file. There are still some parts needed to be corrected before publication.
Q. Missing parts for Figure legend as Figure 1 which might be related to PDF file issue. For a paragraph, it usually consisted of 5-10 sentences. However, there are still some paragraphs consisted of 1 or 2 sentences which might be not good for a reader. Please consider rearrangement such as line 303-315. Please check the paper submitted. Line 298, extra space.
R. I’m really sorry but there was a problem in reformatting the manuscript in pdf. Many mistakes indicated are a consequence of this alteration. As suggested we checked the text and corrected any errors.